# Effects of Temperature Change on the Soil Water Characteristic Curve and a Prediction Model for the Mu Us Bottomland, Northern China

**Xiaoying Qiao [1,2,\*], Shaoyang Ma [1], Guixing Pan [3] and Guanglu Liu [1]**

[1] School of Environmental Science and Engineering, Chang'an University, Xi'an 710054, China; shaoyangma@163.com (S.M.); guanglu_liu@126.com (G.L.)

[2] Key Laboratory of Subsurface Hydrology and Ecological Effect in Arid Region of Ministry of Education, Chang'an University, Xi'an 710054, China

[3] Inner Mongolia Geological Environmental Monitoring Institute, Hohhot 010020, China; guixingpan@163.com

\* Correspondence: qiaoxiaoy@163.com; Tel.: +86-135-7204-6816

**Abstract:** The soil-water characteristic curve (SWCC) is the basis for obtaining the hydraulic conductivity parameters of a soil as well as for using soil water and heat transport models. At present, the curve can be obtained by two methods: by direct measurement and by empirical formula. Direct measurement is both difficult and time-consuming. By contrast, fitting the SWCC with a suitable empirical formula is stable and convenient. The van Genuchten (VG) model has the advantage of universal applicability due to its use of a statistical aperture distribution model for estimating hydraulic conductivity. This study selected the Mu Us Bottomland as a study area. Data on the water content and water potential of undisturbed soil from this site were obtained with a Ku-pF instrument and a self-designed soil column experiment with temperature settings of 13 °C, 18 °C, 23 °C, 27 °C, and 30 °C. The variation of four main parameters in the VG model with temperature was analyzed based on thermodynamic theory and considering the effect of temperature on soil capillary pressure via its effects on surface tension and contact angle. A prediction model for the soil-water characteristic curve of the Mu Us Bottomland was then constructed, and its applicability was further analyzed. The temperature dependence of the SWCC demonstrated here provides an important scientific basis for agricultural production, farmland water conservancy, and the design of soil and water conservation engineering projects.

**Keywords:** soil-water characteristic curve; van Genuchten model; parameter fitting; temperature effect; Mu Us Bottomland

## 1. Introduction

The soil moisture characteristic curve (SWCC) indicates the moisture retention and transport characteristics of a soil. The SWCC is related not only to the soil's intrinsic properties such as its texture and structure [1] but also to the soil temperature [2–5]. Changes in soil temperature and water content can lead to the occurrence of frost heave, bank collapse, masonry cracking, and other diseases in a water conservancy project [6–10]. It is therefore of great theoretical and engineering significance to study the effect of temperature on the SWCC of unsaturated soil.

Researchers have two views with respect to the temperature-dependence of the SWCC. One, based on surface tension–viscous flow theory [11,12], states that a change in temperature affects only the surface tension. For example, Philip and de Vries [13] established a relationship between suction and temperature change based on the Laplace equation, and Wang et al. [14] and Tong [15] derived the effect of temperature on a SWCC model theoretically. However, the results of later experiments

gradually revealed the insufficiency of this view, as they indicated that the effect of temperature on the capillary pressure of unsaturated soil cannot be attributed solely to its effect on surface tension [4,16–21]. Under the conditions of high temperature and low suction, the measured suction value is lower than the value predicted by the model, and the deviation is relatively large, indicating that the influence of temperature on the soil moisture characteristic curve is not limited to its relationship with surface tension [22]. On the other hand, as well as surface tension, temperature change has been found to affect the apparent contact angle. It has been proposed that the relationship between soil water potential and the temperature coefficient can be fitted by an exponential function [16]. Grant and Salehzadeh [23] explored the temperature sensitivities of the capillary pressure functions (CPFs) proposed by Philip and de Vries [13] and proposed a chemical–thermodynamic explanation. The experimental and predicted results tend to be consistent, but quantitative mechanical research on the effect of temperature changes on SWCC has not yet been conducted. It has been shown that the contact angle is affected by the distribution of moisture in silica sand soil and that at a given water content an empirical formula has been derived for predicting the temperature-dependence of the contact angle [20], and experimental research on three different types of soil samples has further validated it [21]. Romero et al. [24] carried out a large number of experimental studies on the variation of the SWCC with temperature and dry density and revised van Genuchten's equation for the SWCC, finding that it is necessary to consider the effect of temperature on the liquid–gas contact angle as well as on the surface tension [25–27].

Past research has generally adopted the method of selecting an appropriate theoretical model and then comparing it with experimental results. Due to limited research, most of the theoretical models are empirically based and lack a rigorous theoretical foundation. The objectives of this study were therefore: (i) to acquire the SWCC of soil at different temperatures by two types of experiment, a soil column experiment and Ku-pF unsaturated hydraulic conductivity measurement; (ii) to analyze changes to the various parameters in the van Genuchten (VG) model that resulted from the effects of temperature on soil capillary pressure via its effect on both the surface tension and the contact angle and then construct a prediction model for the Mu Us bottomland; (iii) to evaluate the SWCC model for unsaturated hydraulic conductivity at different temperatures.

## 2. Materials and Methods

### 2.1. Soil-Water Characteristic Curve Model at Different Temperatures

Since the SWCC cannot be directly derived from theory, scholars have generally used various empirical formulas to describe it such as the Brooks–Corey model [28], Gardner model [29], van Genuchten model [30], Russo model [31], and Fredlund and Xing model [32]. Xu et al. [33] noted that the van Genuchten model has the advantage of universal applicability due to its use of a statistical aperture distribution model for estimating hydraulic conductivity. The coefficient of determination ($R^2$) and the root mean square error (RMSE) are used to evaluate the model fit. The larger the correlation coefficient and the smaller the RMSE error, that greater the fitting accuracy.

Based on the theory of surface tension–viscous flow, temperature changes affect the viscosity, surface tension and density of soil moisture [12], and it thus affects the soil water potential [23] through the relationship expressed by Equation (1):

$$\Psi = \frac{2\sigma^{lg}\cos(\gamma)}{r} \tag{1}$$

where $\Psi$ [Pa] is the capillary pressure; $\sigma^{lg}$ (N·m$^{-1}$) is the surface tension of the liquid–gas interface; $\gamma$ (°) is the contact angle; $r$ (m) is the average radius of the liquid–gas interface.

The partial derivative of $\Psi$ with respect to $T$ from Equation (1) is expressed as:

$$\frac{\partial\Psi}{\partial T} = \frac{\Psi(\theta)}{\sigma^{lg}}\frac{\partial\sigma^{lg}}{\partial T} + \frac{\Psi(\theta)}{\cos(\gamma)}\frac{\partial\cos(\gamma)}{\partial T} \tag{2}$$

The first term on the right-hand side of Equation (2) represents the equation deduced by Philip and de Vries [13], which refers to the influence of temperature on soil water potential through its influence on surface tension [34], whereas the second term stands for the effect of temperature on contact angle [23].

The temperature-dependence of liquid–gas surface tension within the temperature range of −10–50 °C can be closely described with a linear function. Frequently, it is assumed that $\Psi$ is also a linear function of temperature and that, for a given soil water content, $\Psi(\theta,T)$ can be expressed as:

$$\psi(\theta, T) = a(\theta) + b(\theta)T \tag{3}$$

Introducing a soil-specific parameter $\beta_0$, Grant and Salehzadeh [23,35] put forward the formula:

$$\psi(\theta, T) = \psi(\theta, T_r)\left(\frac{\beta_0 + T}{\beta_0 + T_r}\right) \tag{4}$$

where $\Psi(\theta,T)$ is the soil capillary pressure at a given soil water content; $T$ [K] and $T_r$ [K] are the reference and observed temperatures, respectively; $\theta$ is the volumetric water content (dimensionless); $\beta_0$ is a constant empirical coefficient associated with water content and the effect of temperature on capillary pressure.

Bachmann et al. [21] combined the van Genuchten model and the inferred SWCC at different temperatures, list on Equation (4):

$$\theta(\psi, T) = \theta_r + \frac{\theta_s - \theta_r}{\left[1 + |\alpha h(T)|^n\right]^m} \tag{5}$$

where $\theta(\Psi,T)$ is the volumetric water moisture (cm$^3$·cm$^{-3}$); $\theta_s$ is the saturated volumetric water content (cm$^3$·cm$^{-3}$); $\theta_r$ is the residual volumetric water content (cm$^3$·cm$^{-3}$); $h(T)$ is the water pressure head of the linear function, list Equation (3) (cm); $\alpha$[m$^{-1}$] is an empirical fitting parameter related to the air entry value; $m$ and $n$ are curve shape parameters, $m = 1 - 1/n$ or $m = 1 - 1/2n$.

## 2.2. Sample Collection

In-situ samples were collected from the Mu Us Bottomland in the town of Tuke in Wushenqi County, Mongolia Autonomous Region, latitude 38°59′7.73″ and longitude 109°21′18.62″ (Figure 1). The study area has an altitude of 1000–1600 m, annual average temperature variation from 6.0 °C to 8.5 °C, and annual precipitation from 50 mm to 440 mm; there is a large amount of inter-annual variation in precipitation, but about 60–75% generally falls within 7–9 months. Annual evaporation varies from 1800 to 2500 mm. The soil in the study area is a non-isothermal medium and is the result of earlier lakes and shoals in what is now the Mu Us desert, which is characterized by sand dunes surrounding a sandbank basin [36]. The terrain of the Mu Us bottomland is relatively flat, and the soil is mainly aeolian sand. The phreatic aquifer consists mainly of Upper Pleistocene Sjara-Osso-Gol Formation alluvial-lake layers and Holocene-series aeolian sand.

The particle size distribution within the samples was determined using a Bettersize 2000 laser (Dandong Baite Instrument Co., Dandong, China) with a measurement range of 0.02–2000 μm (Figure 2), and we named this material as a silty fine sand.

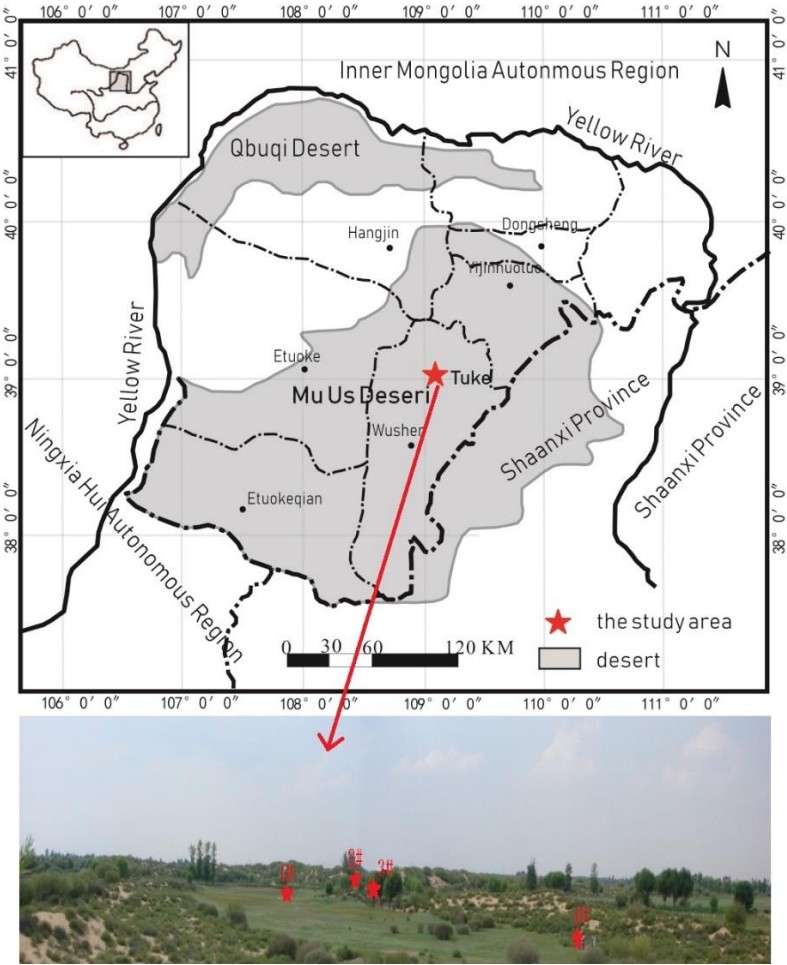

**Figure 1.** The location of the in-situ experiment.

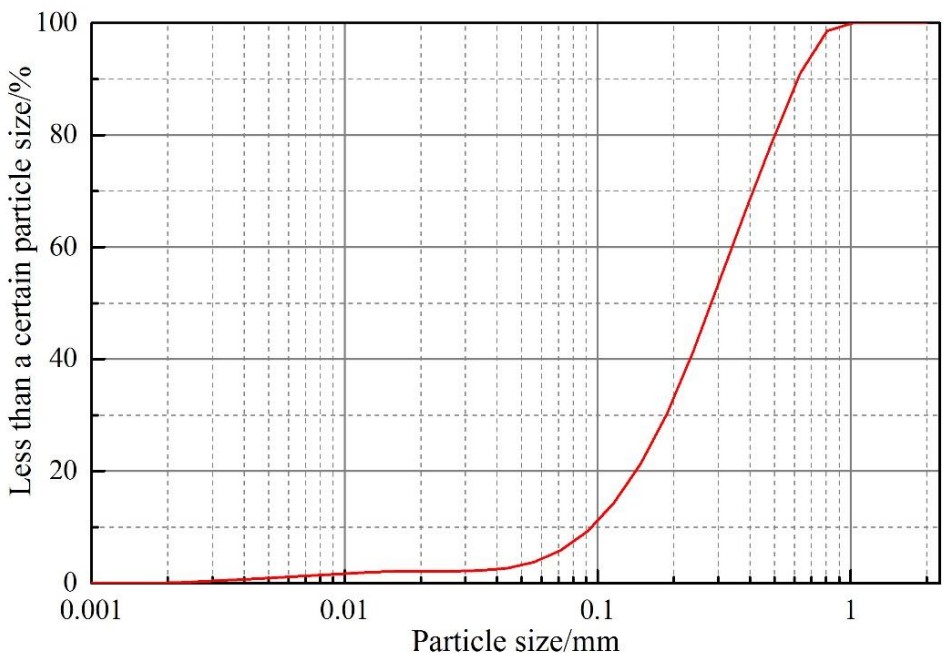

**Figure 2.** Logarithmic probability particle gradation curve.

*2.3. Experimental Setup*

The experiment was conducted using two instrument sets: a Ku-pF unsaturated conductivity measuring instrument and a self-designed soil column experiment.

2.3.1. Soil Column Experiment

The soil column experiment incorporated three pieces of experimental apparatus: an experimental soil column, water supply system, and data collection system. The water potential sensor adopts the research and development patent of Chang'an University, and it was produced by the Baoji Sino-US joint venture Microsensor Co., Ltd (Baoji, China). Its measuring range is −70–20 KPa [37], which consists of three parts: a clay head, a sensor that transforms the pressure signal into an electrical signal, and a connecting rod. The temperature and volumetric water content monitoring system used the 5 TM soil moisture and temperature sensors produced by Decagon Devices, Inc. (2365 NE Hopkins Court, Pullman WA 99163, USA), which have a temperature measurement range of −40–60 °C and resolution of 0.1 °C and a volumetric water content measurement range of 0–100%, accuracy of 1–2%, and resolution of 0.08%. Data collection was adopted with two types of EM50 produced by Decagon Devices, Inc. and a distributed data acquisition automation system produced by Xi'an Yongtai Sensor Technology Limited Company (No. 55, Mingguang Road, Economic and Technological Development Zone, Xi'an). A piezometer tube was used to measure saturated hydraulic conductivity.

The soil column consists of two parts as shown in Figure 3. The upper part is 1.3 m high, the lower part is 1 m high, and the inner diameter is 0.618 m [38]. During the experiment, Figure 3 (9) was filled with soil sample and Figure 3 (12) was filled with filter material (numbers in brackets refer to the components in Figure 3). The soil column temperature was controlled by the temperature of the water supply box. The procedure for the soil column experiment was as follows. First, the soil sample of a certain bulk weight was added to Figure 3 (1). Water was supplied to the soil column and was drained when the soil column had become saturated. This process was repeated 2–3 times. Figure 3 (4) and Figure 3 (5) were installed, and the process was repeated again in order to ensure complete contact between the sensors and the soil. Secondly, Figure 3 (3) and (13) were opened to control the flow rate. The saturated hydraulic conductivity was measured when the reading at Figure 3 (6) had become stable. The input from Figure 3 (3) was changed at intervals of 20 cm successively, and the saturated hydraulic conductivity under different pressure heads Thirdly, Figure 3 (14) was opened to allow the soil to drain freely, and the negative pressure and water content data under the unsaturated state were recorded. Measurement was stopped when the residual moisture content appeared. Finally, the water temperature was controlled to 13 °C, 18 °C, 23 °C, 27 °C, and 30 °C, respectively, and steps 1, 2, and 3 were repeated to obtain the data at different temperatures. Water moisture and capillary pressure changed linearly from the base to the top of the column. The soil water content and water potential were simultaneously measured at different soil depths and 5 min interval data collection. Moreover, to minimize lateral heat fluxes, the sides of the experimental soil column were sealed with three layers of 1 cm-thick extruded polystyrene insulation.

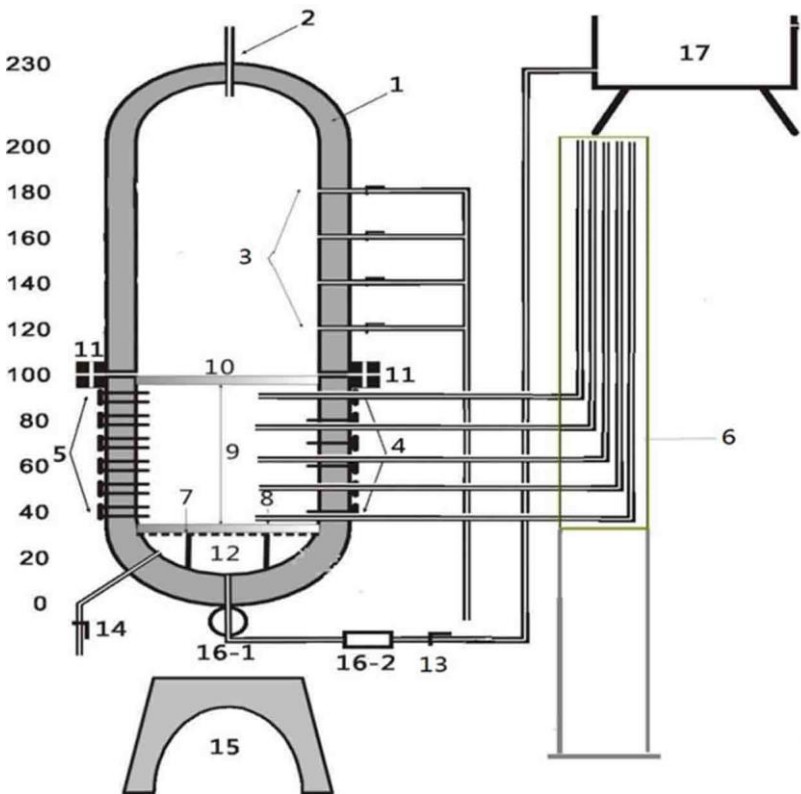

**Figure 3.** The schematic map of the experimental equipment. 1. Soil column; 2. top exhaust pipe; 3. upper water level control drain pipes; 4. water potential sensors; 5. temperature and volumetric water content sensors; 6. piezometer tube; 7. filter plate; 8. filter layer; 9. loading soil sample area; 10. upper filter material; 11. flange plate; 12. cavity; 13. water supply control ball valve; 14. drain control ball valve; 15. pedestal; 16. water meter (16-1), pressure pump (16-2); 17. constant temperature water supply box.

### 2.3.2. Ku-pF Experiment

The Ku-pF unsaturated conductivity measurement instrument is made by UGT, Germany, and can measure the SWCC of undisturbed soil and the unsaturated soil hydraulic conductivity simultaneously. The soil samples were subjected to the targeted temperature using an environmental chamber at 13, 18, 23 °C, respectively. The main experimental procedure was as follows. First, the undisturbed soil samples were placed in shallow water plates for 24 h to saturate them and then sealed basally. Second, the free surface was exposed to evaporation and the gradient of the water movement that ensued was measured and recorded. Up to 10 soil samples can be examined simultaneously with the Ku-pF apparatus because the sample rings are placed on a star-shaped sample-changer and periodically guided across scales at suitable intervals. The volume of water flowing through the sample surface at set intervals was determined on the basis of these weight measurements. The gradient of the soil-moisture tension was calculated in each sample ring by two tensiometers installed 3 cm apart. The readings were recorded synchronously with the weighing cycle. The experiment was considered to finish when the tension-meter reading was at 75–85 kPa; this usually took about 8–9 days. Finally, the dry bulk density was measured after heating in an oven at 105 °C for 12 h. A detailed introduction to the Ku-pF instrument can be found in its operating instructions [39].

## 3. Results and Discussion

### 3.1. Results from the Two Devices without Any Temperature Effect

To be precise, the datum acquired from the test is injected into van-Genuchten model in ORIGIN to obtain SWCC in a wider range of suction. ORIGIN is a scientific drawing and data analysis software developed by OriginLab. It's headquartered in Northampton, Massachusetts, USA. Figure 4 shows the soil column and Ku-pF-derived SWCCs at a normal temperature (18 °C) to provide base data. Obviously using two types of devices over very different suction ranges. To make up for the limitation of heat insultation of soil column experiment, it is necessary to reconcile the results from the two devices when considering the effect of temperature on SWCC.

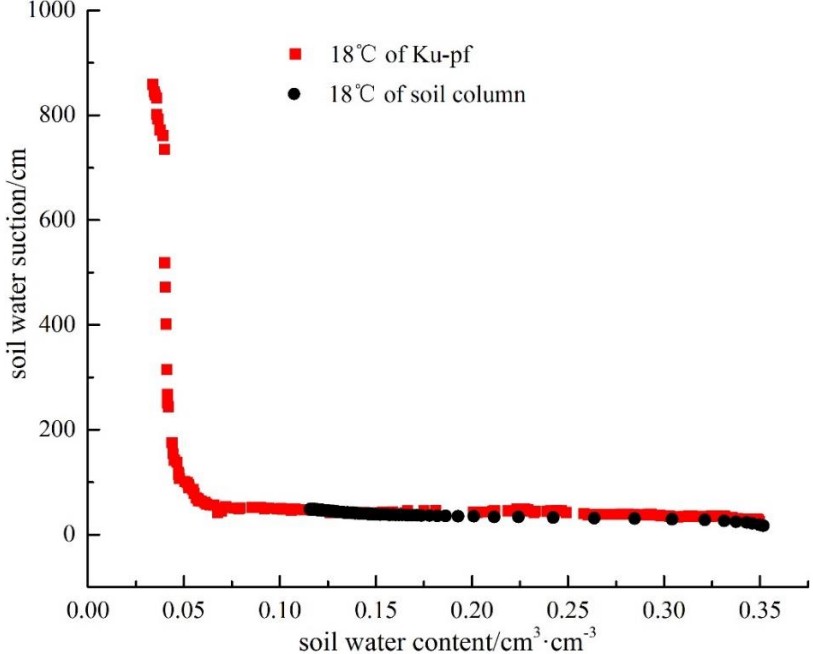

**Figure 4.** The experimental result of two different devices.

### 3.2. Results of Soil Column Experiments at Different Temperatures

Figure 5 shows the SWCCs of soil columns at five temperatures. Average values for the saturated and residual water content as well as parameters $\alpha$ and $n$ at five temperatures are listed in Table 1. Figure 5 shows that the SWCC of the undisturbed soil capillary pressure ranges from 5 cm to 50 cm and thus belongs to the lower suction section. Though Figure 5 apparently shows a good fit, and Table 1 shows good R-squared values, this is a product of the many data points in the 0.10 to 0.20 water content range, where a smooth development of soil water content is to be expected. However, near air entry, the data are less accurate and reflect the true nature of these measurements and curve fitting. The amount of scatter in these data points is of the same magnitude as the temperature effect.

**Table 1.** Fitting parameters of soil-water characteristic curves (SWCC) at different temperatures from the soil column experiment.

| Temperature (°C) | $\theta_r$ | $\theta_s$ | $\alpha$ | $n$ | $R^2$ | RMSE |
|---|---|---|---|---|---|---|
| 13 | 0.1097 | 0.3661 | 0.0313 | 10.65 | 0.9956 | 0.002493 |
| 18 | 0.1143 | 0.351 | 0.0307 | 10.72 | 0.998 | 0.002193 |
| 23 | 0.1077 | 0.3508 | 0.0311 | 9.31 | 0.9953 | 0.002364 |
| 27 | 0.1073 | 0.3437 | 0.0306 | 9.62 | 0.9962 | 0.002365 |
| 30 | 0.1012 | 0.3455 | 0.0312 | 8.189 | 0.9963 | 0.001904 |

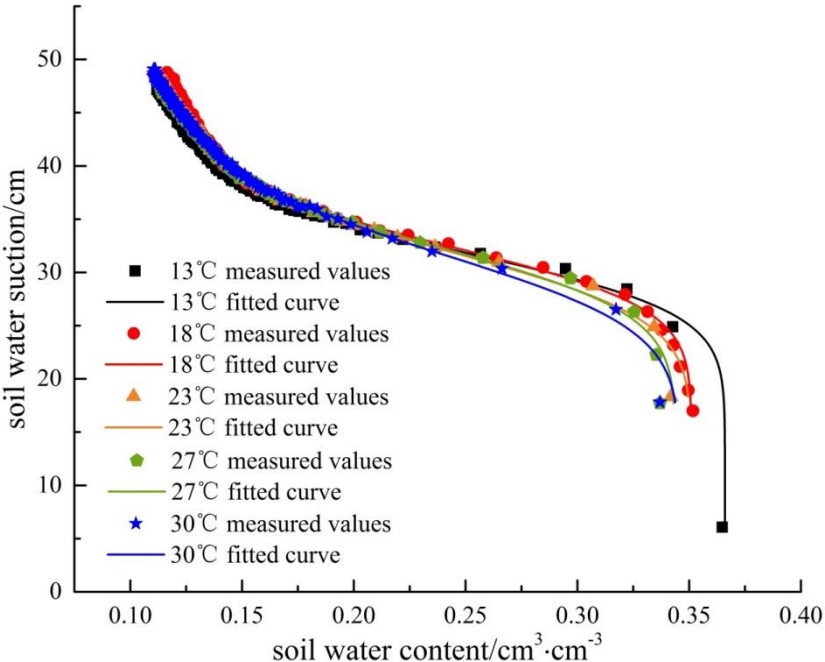

**Figure 5.** Measured data points and fitting curves from the soil column experiment.

### 3.3. Results of Ku-pF Experiments at Different Temperatures

Figure 6 shows that the SWCC of soil capillary pressure ranges 30–900 cm and is therefore within the higher suction section. Average values for the saturated and residual water content as well as parameters $\alpha$ and $n$ at five temperatures are listed in Table 2.

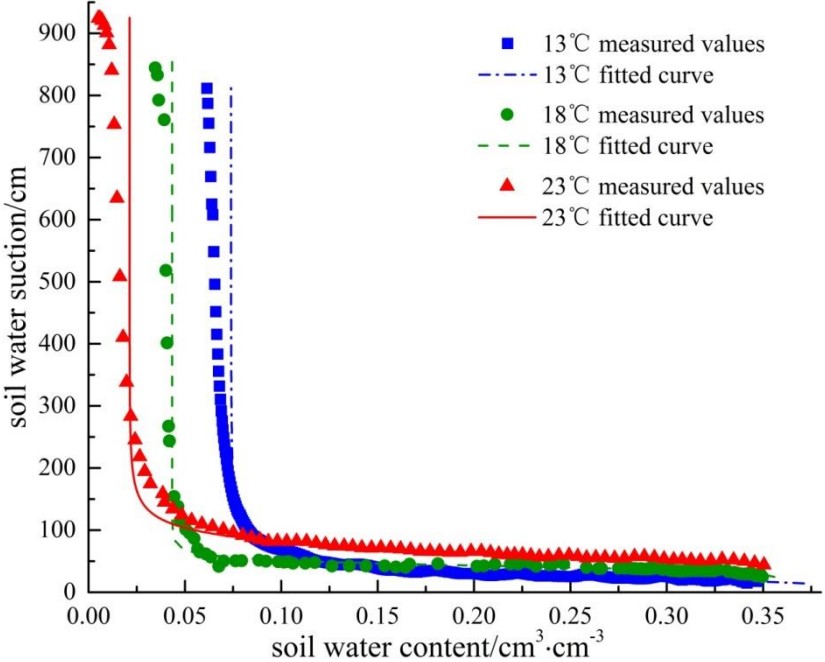

**Figure 6.** Measured data points and fitting curves from the Ku-pF experiment.

**Table 2.** Fitting parameters of soil-water characteristic curve (SWCC) at different temperatures from Ku-pF experiments.

| Temperature (°C) | $\theta_r$ | $\theta_s$ | $\alpha$ | $n$ | $R^2$ | RMSE |
|---|---|---|---|---|---|---|
| 13 | 0.0739 | 0.3993 | 0.0409 | 3.798 | 0.9760 | 0.0139 |
| 18 | 0.0433 | 0.3585 | 0.0237 | 8.736 | 0.9143 | 0.0324 |
| 23 | 0.0219 | 0.4028 | 0.01656 | 5.739 | 0.9861 | 0.0128 |

Together, Figures 5 and 6, Tables 2 and 3 show that the influence of temperature on the SWCC of undisturbed soil is non-negligible. (1) The overall trend of the SWCC was not changed by temperature increase, but there was an obvious change in other stages. The SWCC of undisturbed soil has a tendency to move to the left with an increase in temperature. (2) At a given volumetric water content, the soil water suction increases as the temperature increases. The soil water suction value change is smaller when the volumetric water content is higher and becomes much larger when the volumetric water content is low. (3) A decrease in parameter $\alpha$ indicates a reduction in the water holding capacity of the soil as temperature increases. Schneider and Goss [40] suggest that this is reasonable because of the thick multilayer water film on minerals resembling pure water. The temperature dependences of $\theta_s$ and $\theta_r$ are closely associated with the thermal properties of the liquid–solid interface. Parameter n, a shape parameter corresponding to the effect of the pore size distribution on the slope of the retention curve, increased as temperature rose, demonstrating the effect of temperature change on soil porosity [30,35]. (4) The influence of temperature on the SWCC of undisturbed soil is relatively small in the middle stage but is relatively large in the initial and residual stages. Similarly, Arnfin et al. [41] carried out experiments with a mixture of Calcigel calcium bentonite and quartz sand (50% each) and pure Calcigel calcium bentonite and found that when the water content is constant, the soil suction value is significantly lower at 80 °C than at 20 °C.

**Table 3.** Fitting parameters of the soil-water characteristic curve (SWCC) from the soil column and Ku-pF experiments.

| $T$ (°C) | $\theta_r$ | $\theta_s$ | $\alpha$ | $n$ | $R^2$ | RMSE |
|---|---|---|---|---|---|---|
| 13 | 0.07203 | 0.3919 | 0.03282 | 5.921 | 0.9755 | 0.006541 |
| 18 | 0.0418 | 0.3684 | 0.0287 | 5.815 | 0.9872 | 0.009596 |
| 23 | 0.02118 | 0.3642 | 0.01595 | 5.784 | 0.9829 | 0.01275 |

*3.4. Prediction Model for the Soil-Water Characteristic Curve (SWCC) at Different Temperatures and Error Analysis*

The data from the Ku-pF measurement system and the soil column experiment were combined to acquire the fitting parameters of the SWCC at three temperatures to cover a complete and wide temperature range. These are listed in Table 3.

The relation between the value of *n* and temperature change can be approximately expressed as follows:

$$n = n_0 + \kappa_n(T_m - T_r) \tag{6}$$

Similarly, the saturated volumetric water content and the residual volumetric water content decreased linearly with increasing temperature and so can be approximated with the following formulas.

$$\theta_s = \theta_{s0} + \lambda_s(T_m - T_r) \tag{7}$$

$$\theta_r = \theta_{r0} + \lambda_r(T_m - T_r) \tag{8}$$

$$\alpha = \frac{\alpha_0(\beta + T_r)}{\beta + T_m} \tag{9}$$

where $n_0$, $\theta_{s0}$, $\theta_{r0}$, and $\alpha_0$ are the main parameters for reference temperature $T_r$; $n$, $\theta_s$, and $\theta_r$ are the fitting parameters at a specific temperature, and $\alpha$ is an empirical function related to the air entry value.

From Equation (9), $\beta$ can be inferred as:

$$\beta = \frac{\alpha_0 T_r - \alpha T_m}{\alpha - \alpha_0} \tag{10}$$

Figure 7 shows that the fitting values of parameters $n$ and $\beta$ decrease roughly linearly with an increase in temperature.

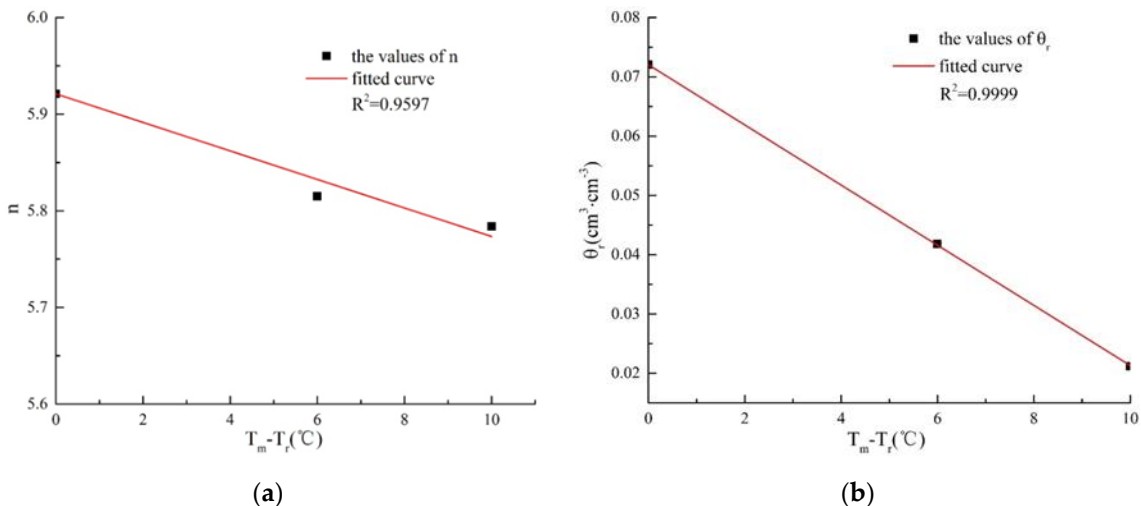

**(a)**                    **(b)**

**Figure 7.** Fitting values for parameters $n$ and $\theta_r$. (**a**) shows the change of n with temperature, and (**b**) shows the change of $\theta_r$ with temperature.

Here selecting 13 °C as the reference temperature and 27 °C as a target temperature, Equations (6)–(8) can be used to obtain fit parameters $n$, $\theta_s$, and $\theta_r$ respectively, and Equation (10) can be used to obtain fit parameter $\beta$. Table 4 shows the relevant fitting parameters.

**Table 4.** Fitting parameters for an soil-water characteristic curve (SWCC) model based on the VG model.

| Parameters | $n_0$ | $\theta_{s0}$ | $\theta_{r0}$ | $\beta$ | $\kappa_n$ | $\lambda_s$ | $\lambda_r$ |
|---|---|---|---|---|---|---|---|
| 13 °C | 5.921 | 0.3919 | 0.072 | 1.41286 | −0.01475 | −0.00307 | −0.00507 |
| $R^2$ | | | | 0.8028 | 0.9597 | 0.9219 | 0.9999 |

Therefore, the prediction model of the SWCC at a random temperature for Mu Us Bottomland eolian sand is as follows:

$$\theta = 0.13794 - 0.00507T_m + \frac{0.27996 + 0.00307T_m}{\left[1 + \left(\frac{0.473h}{1.41286 + T_m}\right)^{6.1128 - 0.01475T_m}\right]^{1 - \frac{1}{6.11275 - 0.01475T_m}}} \tag{11}$$

Figure 8 shows that the predicted values of water potential are basically consistent with the measured values at the same water content. The volumetric water content is close to the actual wetting situation, with the predicted value being a little higher than the measured value and the error range being 0.026 at 27 °C. The volumetric water content is nearly in the dryer soil zone, and the error range is 0.06 at 18 °C.

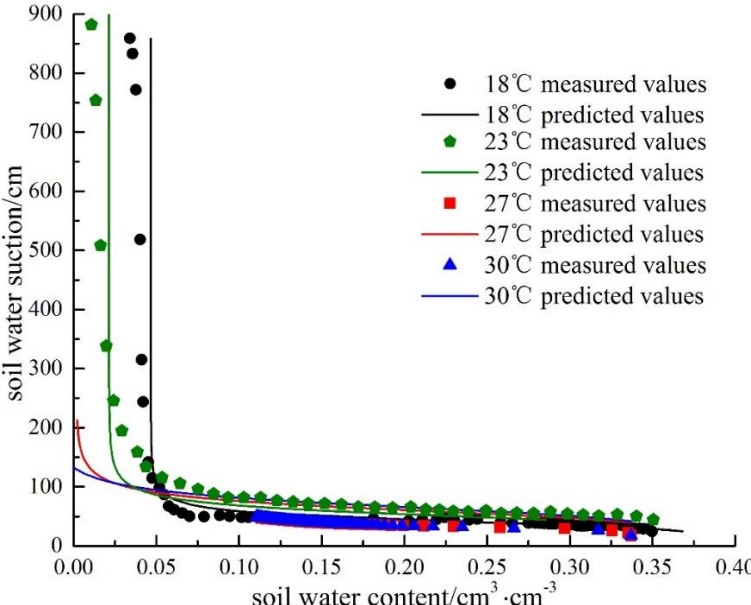

**Figure 8.** Predicted and the measured values for the soil-water characteristic curve (SWCC).

The error analysis to the prediction model of the SWCC considers that temperature is established on the basis of the capillary pressure model, ignoring the effect of temperature on hydration of the particle surface. The interaction energy of a water molecule adsorbed on such a multilayer water film is similar to in adsorption on a pure water surface. The variation in water content with temperature change can be considered as the sum of temperature-dependent changes in the surface tension of the air–water interface and the change in thickness of the double layers in silty fine sand [42,43]. For medium or fine-grained sand, there is less weakly binding water, which can be adsorbed by hydration, in the zone near the saturation water content, so the temperature effect is not obvious. In the near-residual water content or dryer section, not only capillary water but also the weakly binding water, which achieves relatively more adsorption, the temperature effect is thus more obvious, and a SWCC prediction model based on thermodynamic theory is more reasonable. Thus, the effect of temperature on the high-suction section of the SWCC is much larger than on the low-suction section.

In fact, the prediction model of the SWCC for the Mu Us Bottomland has a certain applicable range. A complete SWCC will be obtained in the temperature interval 13–30 °C (here, 13 °C acts as the reference temperature), but only a local section near the saturation water content of SWCC above 30 °C will be acquired, and no predictions can be made for the suction value near the residual water content section due to the limitations on measuring the residual water content.

### 3.5. Effect of Temperature on the Unsaturated Hydraulic Conductivity of Undisturbed Soil

Unsaturated soil water conductivity at different temperatures can be expressed as a function of the water content and capillary pressure of the soil, so the VG model can be further used to estimate its hydraulic conductivity at different temperatures through statistical analysis of the pore size distribution [44]. The relevant expression is as follows:

$$Se(T) = \frac{\theta - \theta_r}{\theta_s - \theta_r} = \left[1 + \left(\alpha |h\,(T)|\right)^n\right]^{-m} \tag{12}$$

$$K(S) = Ks(T)Se^{0.5}\left[1 - \left(1 - Se^{\frac{1}{m}}\right)^m\right]^2 \tag{13}$$

where *Se* is the effective degree of saturation, $0 < Se < 1$, *K(S)* is unsaturated hydraulic conductivity, and *Ks(T)* is saturated hydraulic conductivity at different temperatures. The saturated hydraulic

conductivity of the soil at the pre-designed temperature was measured by the piezometric tube, and then the soil permeability was calculated with Equation (14).

$$k = K\frac{\mu}{\gamma}\bigg|_T \tag{14}$$

where $k$ is the permeability of the soil ($m^2$), $\gamma$ is the bulk weight of water ($10^{-3}$ kN·m$^{-3}$); $\mu$ is the dynamic viscosity coefficient of water ($10^{-6}$ kPa·s$^{-1}$); $T$ is the soil temperature (°C), and $K$ is the saturated hydraulic conductivity (m/d).

The bulk density of water and the dynamic viscosity coefficient of water are known at different temperatures, and, according to Darcy's law, the saturated hydraulic conductivity is 19.6 m/d, 21.3 m/d and 22.9 m/d, respectively, according to the piezometer tube measurements in the column experiment. The unsaturated hydraulic conductivity at 13 °C, 18 °C, and 23 °C can thus be expressed as follows:

$$K(\theta) = 19.6\left[\frac{\theta - 0.07203}{0.31987}\right]^{0.5}\left\{1 - \left[1 - \left(\frac{\theta - 0.07203}{0.31987}\right)^{1.203}\right]^{0.831}\right\}^2 \tag{15}$$

$$K(\theta) = 21.3\left[\frac{\theta - 0.0418}{0.3266}\right]^{0.5}\left\{1 - \left[1 - \left(\frac{\theta - 0.0418}{0.3266}\right)^{1.208}\right]^{0.828}\right\}^2 \tag{16}$$

$$K(\theta) = 22.9\left[\frac{\theta - 0.02118}{0.34302}\right]^{0.5}\left\{1 - \left[1 - \left(\frac{\theta - 0.02118}{0.34302}\right)^{1.209}\right]^{0.827}\right\}^2 \tag{17}$$

Figure 9 shows that at a constant soil moisture content, the predicted unsaturated hydraulic conductivity of the undisturbed soil rises with an increase in temperature, while under the same unsaturated hydraulic conductivity, the soil moisture content increases with decreasing temperature. Moreover, when the water content is low, the effect of temperature on unsaturated hydraulic conductivity is not obvious, but its effect becomes more obvious with an increase in water content. This can also be seen in Table 5.

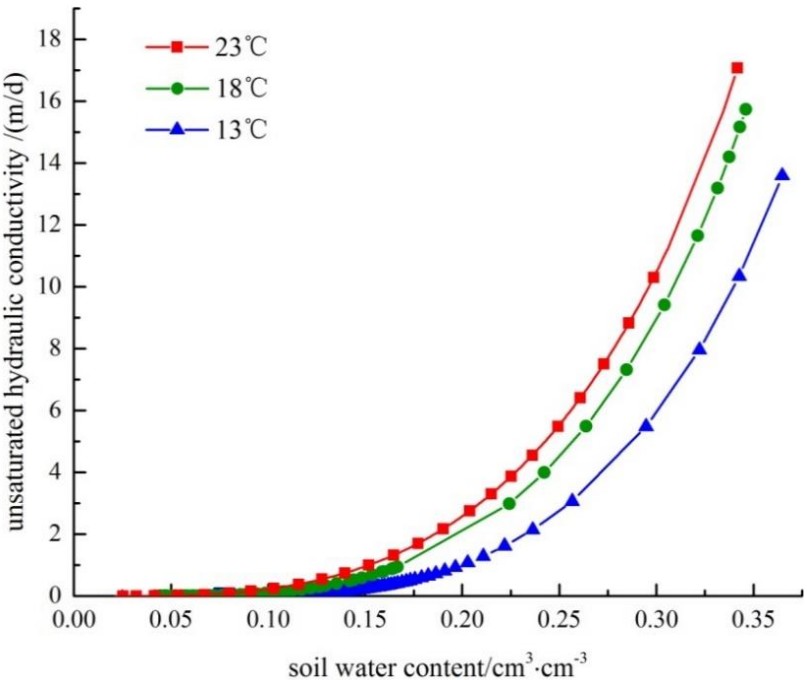

**Figure 9.** Predicted unsaturated soil hydraulic conductivity curves at different temperatures.

**Table 5.** Unsaturated hydraulic conductivity of soil under a 5 °C temperature gradient.

| $\theta$ (%) \ $T$ (°C) | 13 | 18 | 23 | 28 | Scaling Factor |
|---|---|---|---|---|---|
| 0.1 | 0.03 | 0.17 | 0.33 | 0.52 | 0.228 |
| 0.15 | 0.22 | 0.58 | 0.94 | 1.28 | 0.394 |
| 0.2 | 1.62 | 2.98 | 3.68 | 4.05 | 0.559 |
| 0.3 | 5.48 | 9.41 | 10.46 | 10.74 | 0.891 |
| 0.35 | 10.34 | 15.16 | 17.07 | 17.82 | 1.057 |

Following Constantz et al. [44], the temperature-dependence of hydraulic conductivity can be quantitatively predicted from the dynamic viscosity and density of soil water at different temperatures. The effects of temperature on saturated and unsaturated hydraulic conductivity occur by different mechanisms [45]. The former can be attributed the effect of temperature on water viscosity, but the latter results from the effect of simultaneous changes in the soil structure and soil moisture viscosity on the unsaturated hydraulic conductivity of the soil. The effect of temperature change on the unsaturated hydraulic conductivity of soil mainly occurs through three processes: first, it affects the soil water properties, including the surface tension, viscosity, and density of soil water; second, it causes changes in the particle size distribution by affecting the soil structure, resulting in changes in soil structure and porosity; third, the soil moisture content changes with changes in temperature.

## 4. Conclusions

(1) The effect of temperature on the SWCC of undisturbed sand is minor in the middle stage but is relatively much larger in the saturated stage and the residual section. At higher water content, the influence of temperature on the SWCC of silty fine sand is mainly due to its effect on surface tension, whereas it depends on both the surface tension and the contact angle when the water content is lower.

(2) The prediction model of the SWCC at different temperatures based on thermodynamic theory of both the surface tension of soil water and the contact angle avoids the difficulty of measuring the porous radius, as is required for the Grant model [23], and this model thus much more convenient applicability. However, A complete prediction model of SWCC will be obtained in the temperature interval 13–30 °C by the limitation on measuring the residual water content.

(3) The error analysis to the prediction model of the SWCC at different temperatures can been resulted by the sum of temperature-dependent changes in the surface tension of the air water interface and the change in thickness of the double layers in silty fine sand.

(4) When the water content is low, the effect of temperature on unsaturated hydraulic conductivity is not obvious, but its effect becomes more obvious with an increase in water content. The effect of temperature on the saturated hydraulic conductivity and unsaturated hydraulic conductivity of soil provide important practical guidance on moisture transport and the ability to supply water efficiently to plants in the Mu Us bottomland.

**Author Contributions:** Conceptualization, X.Q.; Funding acquisition, X.Q.; methodology, X.Q. and G.P.; validation, S.M.; data curation, S.M. and G.L.; writing—original draft preparation, X.Q. and S.M.; writing—review and editing, G.P. and G.L.

**Acknowledgments:** This research was funded by the National Natural Science Foundation of China (NSFC), grant number 41472222. The authors, therefore, acknowledge NSFC for their financial support with thanks. The authors are also grateful to Yiwei Guo and Chunyan Yang, who collaborated on the experiment.

**Conflicts of Interest:** The authors declare no conflict of interest.

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
