# Peer review of "Effects of Temperature Change on the Soil Water Characteristic Curve and a Prediction Model for the Mu Us Bottomland, Northern China"

_water, doi:10.3390/w11061235_

Round 1

Reviewer 1 Report

Parts of this paper are well done, other parts are not. The introduction is good; it leads to a definite focus on what the paper is about. Some awkward sentences on lines 53-56  is not clear what the authors want to say. Line 59 is odd, research hot-spot is not a familiar or correct term.

The introduction then repeats itself, and there are several places in the paper where the same ideas or statements are repeated. The concepts should be reviewed once, then move onto the rest of the work. Other authors work can be referenced as comparison to results or other direct use, but citing literature about concepts should be in one section in the introduction or brief review of past research. This can also be done on a concept by concept basis. 

Why not put a grain size curve for the sand instead of a table that helps very little. 

Line 105 is not correct. I saw this in the ICP literature and it is incorrect there as well. Go back to the original literature of Decagon Devices and see that they quote a more reasonable accuracy of 1-2% for volumetric water content. 0.08% accuracy is an absurd concept. Why not say the size of the soil specimen being tested in the column? It was not clearly stated how this device is controlled. Do you just let the soil drain/dry with a free draining base in order to achieve unsaturated conditions? You never said how and when you took measurements. It looks as if you could only get a few points for a particular test with six moisture and six water potential sensors. Is the moisture uniform throughout the specimen, or is there linear increase in suction from the base to the top (ie. 0 cm to -60 cm suction). No useful information about the water potential sensors either since a search for the patent came up empty. There was a patent for a similar column, but that says nothing about accuracy, calibration or repeatability of the water potential sensor. This is important since the rest of your measurements rely on these sensors. 

What sort of test is the Ku-pf device axis translation with constant pressure?, transient pressure? Your explanation of the test is too brief and not helpful. 

I would put the theory before the material and methods section. Equation 5 is not well explained but just a repetition of the Bachmann paper. What does the (T) in Eq. 5 mean? It is not possible to comput anything with Eq. 5. 

The decision to combine the test data was not well thought out. Obviously you are using two very different devices over very different suction ranges. It would have been better to reconcile the results from the two devices without any temperatre effects before trying to combine everything with temperature effects thrown in. This problem becomes apparent when you try to fit VG model curves to both sets of data. 

Data from the column experiment (Figure 3) looks okay, but the curve fitting is misleading. You have a good R-squared value because you had so many data points in the 0.10 to 0.20 water content range where everything happens smoothly. But when you get out near air entry, your data is less accurate and reflects the true nature of these measurements and curve fitting. The amount of scatter in these data points is the same magnitude of the temperature effect. I hope you were not trying to hide this fact from the reader. 

You dont tell the reader that Figure 4 curve fitting is done with combined data from Table 3 until after you spend a great deal of time talking about temperature effects. I checked the curves and it took me some time to fit them with your numbers. This is also the section of the paper (Lines 180-200) that you go back and talk about previous research. Some of this is okay since is relates directly to your results, but most of it should have been presented in a review of literature section. 

The section that discusses error analysis is weak since you are trying to analyze three data points with no data that corresponds to theta r and n in your testing. The combined data do not fit well with either the column data or the Ku-pf tests. Look at your data near theta r for all three tests (Figure 4) and tell me how none of the data points match the fitted curve at or near theta r. This means the curve does not predict theta r very well, or the choice of theta r is a guess. 

Figure 6 is even more problematic since it involves data from less than two of your tests

Nothing was mentioned about the piezometer tubes until line 261 and hydraulic conductivity discussion. How did you get unsaturated condutivity from piezometer tube readings in the column?

I believe that Figure 7 means little since there is no measured data presented with the equations given. 

Author Response

Dear Reviewer: 

   We have modified the article according to your Suggestions, and you can view the detailed information in the file "Reviewer 1".

   I am very glad to receive your advice again. 

     Yours respectfully                                                                                               

     Xiaoying Qiao

Reviewer 2 Report

The work presented by Xiao-ying Qiao  et al., deals with the empirical estimation of the Soil-Water Characteristic curve and evaluate the influence of temperature on SWCC. Although the paper shows original data, the structure is well organized, and the aim of the work well presented, there are two minor missing issues:

1)      In the abstract the authors write that  “The temperature dependence of the SWCC demonstrated here provides an important scientific basis for agricultural production, farmland water conservancy, and the design of soil and water conservation engineering projects”, nevertheless these important practical considerations are no longer made explicit in the conclusions. The paragraph “conclusions” should also be reviewed considering what practical results such studies can bring.

2)      In the description of the study area (paragraph 2.1) is missing the geological framework, it might be useful to show the location of the samples on a geological map (Figure 1).

Author Response

Dear Reviewer: 

    We have modified the article according to your Suggestions, and you can view the detailed information in the file "Reviewer 2".

    I am very glad to receive your advice again.

    Yours respectfully

     Xiaoying Qiao

Round 2

Reviewer 1 Report

I would leave out the comment about the Decagon Device measuring to 0.08% volumetric water content, it is misleading. The accuracy of the device is listed as 1-2% volumetric water content when calibrated to a specific soil. 

Did you perform and index tests on the soil? Atterberg limits? Clay content? 

How did you control temperature with the Ku-pF device? You did not say it anywhere in the paper and it is very important to know how you did this. 

I have attached version 2 with just a couple of comments

Author Response

Dear Reviewer: 

    Thank you for your understanding and support, as well as your valuable suggestions. 

    First of all,  with respect to the problem of clay content. Unfortunately, we have not any plasticity measurements (Atterberg limits). However, through particle size analysis we found that the clay content and silt content is no more than  0.44% and 9.25%,respectively. So it is named as silty fine sand, revised from Line 113 of page 4. It is changed the silt fine sand as well, revised from line 260.

    Secondly, with respect to the temperature control problem of Ku-pF equipment. The temperature was controlled by an environmental chamber in the test. Revised from line 161-162.

Your respectfully.

Xiaoying Qiao